# COVID-19 Vaccine Does Not Increase the Risk of Disease Flare-Ups among Patients with Autoimmune and Immune-Mediated Diseases

**DOI:** 10.3390/jpm11121283

**Published:** 2021-12-02

**Authors:** Larisa Pinte, Florentina Negoi, Georgeta Daniela Ionescu, Simona Caraiola, Daniel Vasile Balaban, Camelia Badea, Diana Mazilu, Bianca Dumitrescu, Bogdan Mateescu, Ruxandra Ionescu, Magda Ileana Parvu, Cristian Baicus

**Affiliations:** 1Faculty of Medicine, Carol Davila University of Medicine and Pharmacy, 050474 Bucharest, Romania; scaraiola@yahoo.com (S.C.); vbalaban@yahoo.com (D.V.B.); cameliabadea72@yahoo.com (C.B.); dimazilu@yahoo.com (D.M.); bia1mar@yahoo.com (B.D.); bogmateescu@gmail.com (B.M.); ruxandraionescu1@gmail.com (R.I.); cbaicus@gmail.com (C.B.); 2Internal Medicine, Rheumatology and Gastroenterology Departments, Colentina Clinical Hospital, 072202 Bucharest, Romania; flori.negoi28@gmail.com (F.N.); ionescu_georgiana93@yahoo.com (G.D.I.); parvumagda@yahoo.com (M.I.P.); 3Gastroenterology and Internal Medicine Departments, Dr. Carol Davila Central Military Emergency University Hospital, 01082 Bucharest, Romania; 4Rheumatology and Internal Medicine Departments, Sf. Maria Clinical Hospital, 011172 Bucharest, Romania; 5Rheumatology Department, Ion Stoia Clinical Centre of Rheumatic Diseases, 030167 Bucharest, Romania

**Keywords:** COVID-19, SARS-CoV-2, vaccine, autoimmune, immune-mediated diseases, flare-up, disease activity, safety

## Abstract

Background: Reports describing post-vaccine autoimmune phenomena, in previously healthy individuals, increased the concerns regarding the risk of disease flare-ups in patients with immune diseases. We aimed to assess the potential risk of disease flare-up, after receiving the COVID-19 (Coronavirus disease 2019) vaccine, during a follow-up period of 6 months. Methods: We performed a prospective cohort study, enrolling the patients with autoimmune- and immune-mediated diseases who voluntarily completed our questionnaire, both online and during hospital evaluations. Based on their decision to receive the vaccine, the patients were divided into two groups (vaccinated and non-vaccinated). Participants who chose not to receive the vaccine served as a control group in terms of flare-ups. Results: A total of 623 patients, 416 vaccinated and 207 non-vaccinated, were included in the study during hospital evaluations (222/623) and after online (401/623) enrolment. There was no difference concerning the risk of flare-up between vaccinated and non-vaccinated patients (1.16, versus 1.72 flare-ups/100 patients-months, *p* = 0.245). The flare-ups were associated with having more than one immune disease, and with a previous flare-up during the past year. Conclusions: We did not find an increased risk of flare-up following COVID-19 vaccination in patients with autoimmune-/immune-mediated diseases, after a median follow-up of 5.9 months. According to our results, there should not be an obvious reason for vaccine hesitancy among this category of patients.

## 1. Introduction

The last recommendations published by The European League Against Rheumatism (EULAR) support vaccination in patients with autoimmune and immune-mediated diseases (AID-IMD) during the quiescent phase of the disease and before starting immunosuppressive treatment if possible [1]. It is well known that the rate of vaccination is suboptimal in these patients, attributed to low rates of referral for vaccination [2] and concerns related to disease flare-up. Until now, epidemiological studies did not support the hypothesis that vaccines may reactivate or may induce AID-IMD [3]. There were reported a few cases of Guillain–Barre syndrome after Influenza vaccine and rare cases of systemic lupus erythematosus exacerbation after papilloma vaccination [4,5]. Due to the risk of COVID-19 (Coronavirus disease 2019) infection, AID-IMD patients were included in the second line of vaccination. However, this decision caused concerns regarding the COVID-19 vaccine’s safety and its effect on disease activity. After the vaccination begun in a large population, case series of post-vaccine immune phenomena were published [6,7]. Disease flare-up in patients with AID-IMD in clinical remission was reported [8,9]. Providing reliable study results concerning this subject is important for future research prospects.

The aim of the study was to assess the potential risk of disease flare-up in AID-IMD patients after receiving the COVID-19 vaccine, during a follow-up period of 6 months.

## 2. Materials and Methods

We performed a prospective cohort study, enrolling the patients on the basis of a questionnaire handled both online, via a Survey Monkey link posted on social media patient groups, as well as in person during regular face-to-face evaluations, and including all consecutive patients presenting to four participant tertiary care centers, from 5 February until 7 May 2021. The questions gathered data regarding demographics, disease activity and treatment, as well as details on post-vaccine adverse events and flare-ups. Subjects were asked to provide their telephone number as ID and follow-up contact information. We adhered to Strengthening the Reporting of Observational Studies in Epidemiology reporting guidelines [10]. The study was conducted in accordance with the General Data Protection Regulation of the European Union, applied from 25 May 2018, and approved by the Ethics committee of Colentina Clinical Hospital (3/08.02.2021). Filling out of the questionnaire was conducted after reading the information page on the survey, which implied the informed consent for the inclusion in the study. The flow diagram of the study is presented in Figure 1.

### 2.1. Participants

Based on their decision to receive the vaccine, the patients were divided into two groups (vaccinated and non-vaccinated). The same questions were addressed to both groups as they were monitored by telephone every 2 months for increases in their disease activity and vaccine-related adverse effects. Participants who chose not to receive the vaccine served as a control group in terms of flare-ups. The patients who decided to receive the vaccine during the follow-up period were included in the vaccinated group, beginning with the vaccination time (first dose). The follow-up period started from the date of the first dose (for the vaccinated group) or from enrolment (for the non-vaccinated group), and lasted until the flare-up was reported or, in the absence of flare-up, until the last evaluation.

We considered the patients to be lost to follow-up if they: (a) withdrew their consent to participate in the research, or (b) did not respond to two consecutive telephone visits including the last one.

The primary outcome measure was the time to first flare, while secondary outcomes included incidence of flare-ups between the two groups (vaccinated and non-vaccinated), as well as vaccine-related adverse effects.

A flare-up was defined as typical symptoms of increase in disease activity that were associated with one of the following: (a) hospital admission, (b) inflammation and biological indicators, and/or (c) treatment adjustments (increase in dose, frequency, or type of medication).

We also documented the timespan from the diagnosis of the immune pathology until enrolment as well as flare-ups reported in the year prior to vaccination/enrolment.

### 2.2. Data Sources/Measurement

The same questions were addressed to both groups (vaccinated and non-vaccinated). Flare-up prior to vaccination or enrolment was measured based on the date provided by the patients upon enrolment. The clinical and biological features evaluated in order to establish an increase in disease activity were addressed based on the disease activity scales currently used for each pathology, and were included in the study as a Appendix A.

### 2.3. Sample Size

With an estimated flare-up incidence in the control group of 5%, we calculated a sample size of 868 patients able to detect a difference to an incidence of at least twice that in the vaccinated group, with an alpha error of 5% and a beta error of 20%.

### 2.4. Data Analysis

Variables with a non-Gaussian distribution were reported as median (min, max) and assessed using nonparametric tests (Mann–Whitney U test). The nominal variables were reported as number (%), and compared by X2 test, or Fisher’s exact test when appropriate. The time until flare-up in the two groups was compared by the Log Rank test in bivariate analysis, and by the Cox model in multivariable analysis. In the multivariable model, the variables associated with flare-ups with a *p*-value ≤ 0.10 were introduced, and the regression was ruled stepwise and backward in order to highlight the potential suppressors [11]. Microsoft Excel 2018 (Microsoft Corporation, USA) and IBM SPSS Statistics for Windows, version 20 (IBM Corp., Armonk, NY, USA) were used. *p* values <0.05 were considered statistically significant.

## 3. Results

A total of 651 surveys were considered potentially eligible, 232 gathered during medical visits and 419 collected online. After eligibility assessment, a total of 623 patients, 416 vaccinated and 207 non-vaccinated, were included in the study.

The vaccinated group consisted mostly of patients enrolled online (299, 71.9%), the majority being women (339/416, 81.5%), with a median age of 50 years (21, 88). The non-vaccinated group included mostly patients enrolled during hospital evaluations, mostly women (183/207, 88.4%), with a median age of 48 years (19, 73).

The types of COVID-19 vaccines administered were Comirnaty (BioNTech/Pfizer, Germany, Mainz) (86%), Vaxzevria (Oxford/AstraZeneca, Nijmegen, Netherlands) (9%), Spikevax (Moderna Biotech, Cambridge, USA) (3%), and Janssen vaccine (Johnson & Johnson, Leiden, Netherlands) (2%).

There was no difference between the two groups in terms of age, comorbidities, number of autoimmune diseases associated, and years from disease diagnosis or disease flare-up during the year prior vaccination/enrolment. Having an autoimmune rheumatic and musculoskeletal disease (AIRD), or taking corticosteroids, synthetic disease-modifying anti-rheumatic drugs (DMARDs) or azathioprine decreased the immunization rate.

More than half (250, 60.7%) of the participants had already received the first dose before enrolling in the study.

The distribution of the variables in the flare-up and non-flare-up groups as well in the vaccinated/non-vaccinated groups are presented in Table 1.

Top symptoms after vaccine administration included pain at injection site, fatigue, headache, chills, joint pain, and myalgia. Overall, no specific difference between AIRD and non-AIRD patients was observed in terms of adverse events. Apart from one patient who developed anaphylactic shock after the second dose, no other life-threatening events were observed. The reported adverse effects were self-limiting and none of them required hospitalization or a change in the immunomodulatory therapy. The top symptoms reported after each dose are listed in Table 2.

A total of 31 participants (7.5%) reported stopping their background medication before vaccination in order to optimize vaccine immunogenicity. There was no difference concerning the risk of flare-up between the patients who stopped their treatment, and those who did not (4/31 versus 21/385, *p* = 0.105).

The median follow-up time was 180 days (min = 8, max = 246): 187 days (8, 246) in vaccinated patients, versus 170 days (16, 237) in non-vaccinated patients, *p* < 0.001.

During the follow-up period, a total of 42 flare-ups were reported, 25/416 (6%) in the vaccinated group and 17/207 (8%) in the non-vaccinated group, *p* = 0.302.

In addition, there was no significant difference concerning the types of COVID-19 vaccines administrated in terms of flare-up development: Comirnaty (BioNTech/Pfizer) (19/359, 5%), Vaxzevria (Oxford/AstraZeneca) (4/37, 11%) or Spikevax (Moderna Biotech) (2/12, 17%), *p* = 0.43.

Three of the 25 flare-ups were reported after receiving the first dose, while the rest developed after the second dose. The three flare-ups reported after the first dose started on days 8, 10 and 12, respectively. Furthermore, 16% (4/25) of the flare-ups started on day 2, 16% (4/25) between days 14 and 21 and 56% (14/25) 21 days after the second dose. The length of flare-up was <7 days in 30% of the patients, 8–21 days in 22% of the patients and >30 days in 48% of the patients. However, there was no difference regarding the median length of flare-up between the two groups: 30 days (4, 149) in the vaccinated and 27.5 days (14, 164) in the non-vaccinated patients, *p* = 0.803.

The incidence-densities of the flare-ups in the vaccinated and non-vaccinated groups were 1.16 and 1.72/100 patients-months, respectively (*p* = 0.245, Log Rank test) (Figure 2).

In bivariate analysis, the flare-ups were associated with having more than one immune disease, taking corticosteroids, and with a previous flare-up during the past year (Table 1). In the Cox model, only having a flare-up during the previous year was associated with a flare-up (Table 3).

## 4. Discussion

Reports describing post-vaccine autoimmune phenomena, such as Guillain–Barre syndrome, thrombotic thrombocytopenia, myocarditis and hyper-inflammatory pathologies [12,13], in previously healthy individuals, increased the concerns regarding the risk of disease flare-up development.

Since pilot studies that allowed the marketing authorization for Spikevax (Moderna Biotech) and Janssen COVID-19 vaccines excluded immunosuppressed subjects, and only 0.3% of the participants in the Comirnaty (BioNTech/Pfizer)clinical trials had rheumatic diseases [14], official data regarding vaccine safety was lacking.

Despite all concerns, receiving the COVID-19 vaccine did not increase the risk of flare-ups among patients with autoimmune-/immune-mediated diseases, according to our results.

We did not find an increased risk of flare-up following COVID-19 vaccination in AID-IMD patients, after a median follow-up of 5.9 months (180 days).

In addition, various studies including the preliminary data from the EULAR COVID-19 Vaccination Registry [15] showed an acceptable safety profile for COVID-19 vaccines, showing no evidence of significant disease flare-ups across different AID-IMD [16,17,18,19].

Since the association between vaccination and flare-ups was not significant, we looked for a possible suppressor variable which should have been associated simultaneously with the outcome (flare), and the risk factor (vaccination), and only corticotherapy met this criterion. We adjusted for corticotherapy (both as a quantitative variable (dosage) and as a nominal variable (corticotherapy yes/no)), but this did not affect the relation between vaccination and flare. We adjusted also for the other potential confounders, which were associated with flare-up with a *p* ≤ 0.10 (Table 1). In fact, corticotherapy was not associated anymore with flare-ups, after we adjusted for having a flare-up in the previous year.

However, vaccine-related decisions regarding background medication adjustments should be approached from a risk–benefit perspective.

The main strength of the study is that it evaluates vaccine safety in terms of flare-ups among AID-IMD patients comparing the vaccinated and non-vaccinated AID-IMD individuals. As we know, the previous studies only presented the incidences of flare-ups among the vaccinated patients, while this is the first study including a control cohort (patients not vaccinated).

Another strength is the medium-term follow-up (median, 5.9 months). Nevertheless, based on the results published until now, it appears that most of the flare-ups usually occurred in the first week following vaccine administration [20,21]; as such, post-vaccination long-term follow-up might not be required. The fact that almost half of the flare-ups reported in our study developed 21 days after vaccine administration raises the possibility that not all of them were vaccine associated.

This study is limited by the observational design and the convenience heterogeneous sample obtained by enrolling patients with a wide spectrum of autoimmune /immune-mediated diseases. Several systemic autoimmune diseases, such as systemic lupus erythematosus, systemic sclerosis and autoimmune thyroid disease, are included statistically less often in the group of vaccinated patients, but this only means that, for a number of diseases, the number of exposed patients is larger than the number of non-exposed, a fact which does not influence the results. A real limitation of the study is that it has no statistical power to draw conclusions for the diseases that had a small number of included patients, but results could be drawn for the autoimmune and immune-mediated diseases in general, which was the objective of this study.

There were more vaccinated patients among those recruited by internet (possibly because they might have been younger, and more educated/high internet users), but the incidence of flare-ups was the same among all the patients, regardless the mean of recruitment. In order to be a confounding factor, an independent variable has to be associated with the outcome (here, the flare). In our study, neither the vaccination status, nor the gender or the enrolment mean were associated with the outcome in bivariate analysis, so we did not need to adjust for gender or enrolment mean in the multivariable analysis.

Furthermore, most of the patients received the Comirnaty (BioNTech/Pfizer) vaccine; as such, a comparative analysis between vaccine types could not be performed.

Another limitation of the study is that flare-ups were self-reported, and the flare-up-related data were not objectively presented by healthcare providers. However, in order to avoid differential misclassification, telephone interviews were performed by specially trained staff who addressed the same follow-up questions to all participants and classified the flare–ups based on the objective criteria previously described in the methods section (the Disease activity evaluation chart is available in the Appendix A).

## 5. Conclusions

We did not find an increased risk of flare-up following COVID-19 vaccination in patients with autoimmune-/immune-mediated diseases, after a median follow-up of 5.9 months.

According to our results, there should not be an obvious reason for vaccine hesitancy among patients with autoimmune and immune-mediated diseases.

## Figures and Tables

**Figure 1 jpm-11-01283-f001:**
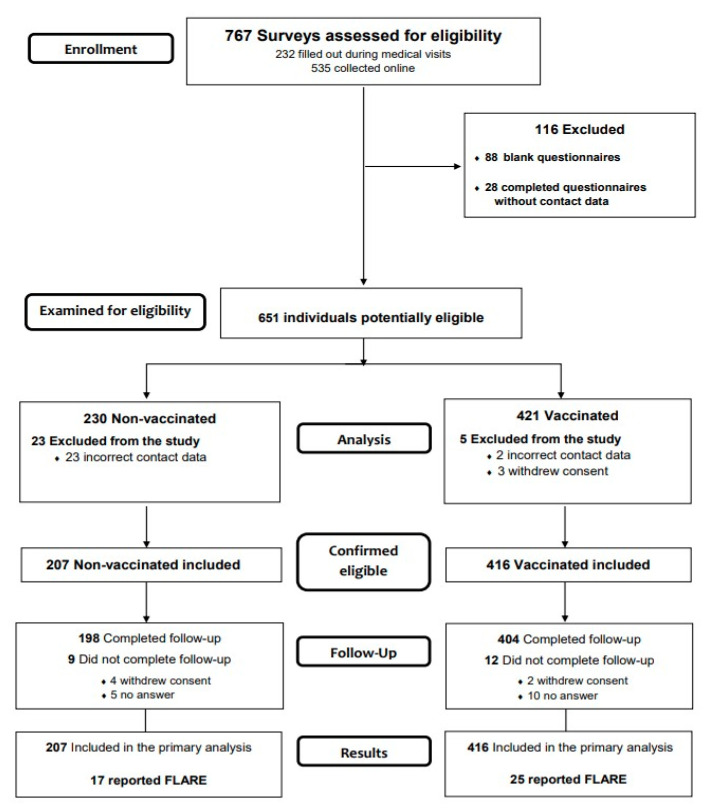
The flow diagram of the study describing the methods: participant’s recruitment, inclusions, exclusions and follow-up.

**Figure 2 jpm-11-01283-f002:**
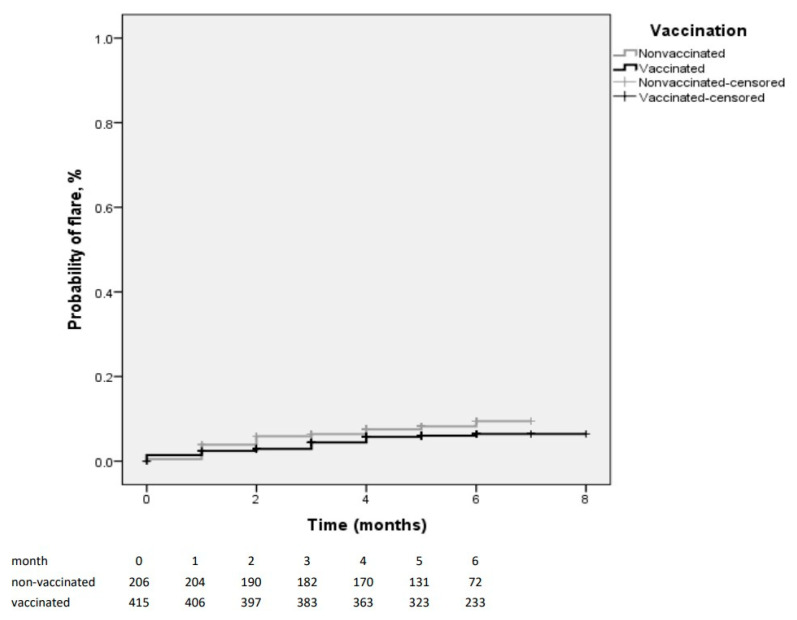
The Kaplan–Meier curves for the flare-ups in the vaccinated and non-vaccinated groups.

**Table 1 jpm-11-01283-t001:** The distribution of the variables in the flare-up and non-flare-up groups as well in the vaccinated/non-vaccinated groups.

	Patients with and without Flare				Vaccinated and Non-Vaccinated Patients			
	Flare-Up (*n* = 42)	Non-Flare-Up (*n* = 581)	*p* Value	Missing	Vaccinated (*n* = 416)	Non-Vaccinated (*n* = 207)	*p* Value	Missing
Enrolled online	31 (73.8%)	30 (63.7%)	0.243		299 (71.9%)	102 (49.3%)	**<0.001**	
Vaccinared	25 (59.5%)	391 (67.3%)	0.312		-	-	-	
Gender (F)	37 (88.1%)	485 (83.5%)	0.522		339 (81.5%)	183 (88.4%)	**0.028**	
Age (y)	46 (21, 83)	49 (19, 88)	0.126		50 (21, 88)	48 (19, 73)	0.290	
Immune disease and comorbidities								
Charlson index	2 (0, 6)	1 (0, 10)	0.512		1 (0, 9)	1 (0, 10)	0.354	
More than one immune disease	16 (38.1%)	118 (20.3%)	**0.011**		83 (20%)	51 (24.6%)	0.180	
Number of immune diseases/patient	1 (1, 4)	1 (1, 6)	**0.006**		1 (1, 4)	1 (1, 6)	0.142	
Immune pulmonary involvement	6 (14.3%)	59 (10.2%)	0.429		40 (9.6%)	25 (12.1%)	0.404	
AIRD ^1^	26 (61.9%)	369 (63.5%)	0.869		239 (57.5%)	156 (75.4%)	**<0.001**	
Rheumatoid arthritis	7 (16.7%)	91 (15.7%)	0.827		59 (14.2%)	39 (18.8%)	0.161	
Systemic lupus erythematosus	9 (21.4%)	88 (15.1%)	0.273		48 (11.5%)	49 (23.7%)	**<0.001**	
Sjögren’s syndrome/Sicca ^2^	7 (16.7%)	71 (12.2%)	0.466		50 (12%)	28 (13.5%)	0.608	
Ankylosing spondylitis	2 (4.8%)	63 (10.8%)	0.298		48 (11.5%)	17 (8.2%)	0.214	
Psoriatic arthritis/psoriasis	4 (9.5%)	49 (8.4%)	0.774		32 (7.7%)	21 (10.1%)	0.360	
Systemic sclerosis/limited scleroderma	1 (2.4%)	30 (5.2%)	0.714		14 (3.4%)	17 (8.2%)	**0.011**	
Antiphospholipid syndrome	3 (7.1%)	22 (3.8%)	0.234		12 (2.9%)	13 (6.3%)	0.051	
Systemic vasculitis	2 (4.8%)	15 (2.6%)	0.320		8 (1.9%)	9 (4.3%)	0.114	
Other AIRD ^3^	1 (2.4%)	14 (2.4%)	1		9 (2.2%)	6 (2.9%)	0.586	
Non-AIRD	16 (38.1%)	212 (36.5%)	0.869		177 (42.5%)	51 (24.6%)	**<0.001**	
Inflammatory bowel disease	5 (11.9%)	39 (6.7%)	0.207		34 (8.2%)	10 (4.8%)	0.138	
Celiac disease	1 (2.4%)	18 (3.1%)	1		14 (3.4%)	5 (2.4%)	0.626	
Primary biliary cholangitis	1 (2.4%)	8 (1.4%)	0.469		4 (1%)	5 (2.4%)	0.167	
Autoimmune hepatitis	0 (0%)	11 (1.9%)	1		8 (1.9%)	3 (1.4%)	1	
Myasthenia gravis	6 (14.3%)	22 (3.8%)	**0.008**		15 (3.6%)	13 (6.3%)	0.151	
Multiple sclerosis	1 (2.4%)	24 (4.1%)	1		17 (4.1%)	8 (3.9%)	1	
Hematological diseases ^4^	1 (2.4%)	4 (0.7%)	0.295		4 (1%)	1 (0.5%)	1	
Cutaneous diseases ^5^	1 (2.4%)	16 (2.8%)	1		12 (2.9%)	5 (2.4%)	1	
Autoimmune thyroid disease	9 (21.4%)	128 (22%)	1		109 (26.2%)	28 (13.5%)	**<0.001**	
Other non-AIRD ^6^	6 (14.3%)	44 (6.7%)	0.136		40 (9.6%)	10 (4.8%)	0.042	
Treatment								
Corticosteroids	16 (38.1%)	120 (20.7%)	**0.012**		71 (17.1%)	65 (31.4%)	**<0.001**	
Corticosteroid dose (mg)	10 (5, 45)	10 (2,5, 40)	**0.015**	29	6.25 (5, 20)	10 (3, 45)	**<0.001**	29
Synthetic DMARDs ^7^	12 (28.6%)	226 (38.9%)	0.194		143 (34.4%)	95 (45.9%)	**0.007**	
Hydroxychloroquine	8 (19%)	125 (21.5%)	0.846		81 (19.5%)	52 (25.1%)	0.119	
Methotrexate	2 (4.8%)	64 (11%)	0.298		42 (10.1%)	24 (11.6%)	0.582	
Sulfasalazine	2 (4.8%)	48 (8.3%)	0.566		34 (8.2%)	16 (7.7%)	1	
Leflunomide	1 (2.4%)	20 (3.4%)	1		10 (2.4%)	11 (5.3%)	0.096	
Biologic DMARDs ^7^	10 (23.8%)	119 (20.6%)	0.561		89 (21.4%)	40 (19.4%)	0.600	
Mycophenolate mofetil	1 (2.4%)	17 (2.9%)	1		13 (3.1%)	5 (2.4%)	0.801	
Azathioprine	5 (11.9%)	40 (6.9%)	0.216		23 (5.5%)	22 (10.6%)	**0.031**	
Flare-up-related variables								
Disease diagnosis-enrollement/vaccine (years)	11 (1, 40)	9 (0, 56)	0.608	29	9 (0, 50)	10 (0, 56)	0.999	29
Flare-up in the year prior enrolement/vaccination	22 (73.3%)	189 (47.4%)	**0.007**	194	130 (46.8%)	81 (53.6%)	0.189	194
Enrollment/vaccination-flare-up timelaps (months)	-	-	-		6.13 (0, 8)	5.57 (1, 8)	**<0.001**	
COVID-19 ^8^ prior enrollement/vaccine	4 (9.5%)	81 (13.9%)	0.640		75 (18%)	10 (4.8%)	**<0.001**	
Adverse events after dose 1	24 (92.3%)	283 (72.4%)	0.022		307/416	-	-	
Adverse events after dose 2	21 (80.8%)	252 (64.5%)	0.134		273/416	-	-	
Treatment adjustment before vaccination	4 (16%)	27 (6.9%)	0.105		31/416	-	-	
Treatment adjustment before flare	3/42 (7.1%)	-	-		1 (4%)	2 (11.8%)	0.556	
Average length of flare-up (days)	30 (4, 164)	-	-	4	30 (4, 149)	27.5 (14, 164)	0.803	4
Still having flare-up at the last assessment	19 (45.2%)	18 (3.2%)	**<0.001**	21	18 (4.5%)	19 (9.6%)	**0.018**	21
Flare-up management								
Hospitalization	15/42	-	-		10/25	5/17	-	
Treatment adjustment	36/42	-	-		21/25	15/17	-	

^1^ AIRD—autoimmune rheumatic diseases; ^2^ Sicca-Sicca syndrome (xerostomia, xeroftalmia); ^3^ Other AIRD—dermatomyositis/polymyositis and mixed connective tissue disease; ^4^ Hematological diseases—hemolytic anemia, paroxysmal nocturnal hemoglobinuria, idiopathic thrombocytopenia; ^5^ Cutaneous diseases—vitiligo, cutaneous lupus, cutaneous vasculitis, pemphigus; ^6^ Other non AIRD—sarcoidosis, type-1 diabetes mellitus, hyper IgD syndrome; ^7^ DMARDs-disease-modifying anti-rheumatic drugs; ^8^ COVID-19-Coronavirus disease 2019. Statistical significant values are marked with Bold.

**Table 2 jpm-11-01283-t002:** Top symptoms reported after the 1st and 2nd COVID-19 vaccine doses.

	1st Dose (*n* = 416)				2nd Dose (*n* = 407)			
	Non-AIRD ^1^ (*n* = 134, 75.7%)	AIRD (*n* = 173, 72.4%)	*p* Value (0.499)	Total Adverse Events (*n* = 307)	Non-AIRD (*n* = 125, 70.6%)	AIRD (*n* = 148, 61.9%)	*p* Value (0.076)	Total Adverse Events (*n* = 273)
Pain in injection site	81 (45.3%)	99 (41.4%)	0.485	180	86 (48%)	82 (34.3%)	0.005	168
Local swelling	9 (5%)	6 (2.5%)	0.192	15	6 (3.4%)	4 (1.7%)	0.338	10
Local redness	22 (12.3%)	23 (9.6%)	0.427	45	8 (4.5%)	7 (2.9%)	0.435	15
Fatigue	46 (25.7%)	51 (21.3%)	0.349	97	41 (22.9%)	52 (21.8%)	0.813	93
Headache	33 (18.4%)	44 (18.4%)	1	77	24 (13.4%)	25 (10.5%)	0.361	49
Myalgia	24 (13.4%)	35 (14.6)	0.777	59	25 (14%)	25 (10.5%)	0.289	50
New/increased joint pain	25 (14%)	35 (14.7%)	0.888	60	10 (5.6%)	20 (8.4%)	0.340	30
Chills	24 (13.4%)	35 (14.6%)	0.777	59	28 (15.6%)	34 (14.5%)	0.782	62
Fever	8 (4.5%)	13 (5.4%)	0.822	21	22 (12.3%)	24 (10%)	0.528	46
Allergic reactions	3 (1.7%)	2 (0.8%)	0.655	5	1 (0.6%)	0 (0%)	0.427	1
Skin rash	3 (1.7%)	3 (1.3%)	1	6	1 (0.6%)	0(0%)	0.428	1
Anaphylactic shock	0 (0%)	0 (0%)	0	0	0 (0%)	1 (0.4%)	1	1
Lymphadenopathy	4 (2.2%)	5 (2.1%)	1	9	5 (2.9%)	4 (1.7%)	0.501	9
Paresthesia	1 (0.6%)	6 (2.5%)	0.247	7	2 (1.2%)	5 (2.1%)	0.704	7
Digestive symptoms	4 (2.2%)	5 (2.1%)	1	9	7 (4%)	12 (5%)	0.813	19
Dizziness	2 (1.1%)	7 (2.9%)	0.311	9	6 (3.5%)	6(2.5%)	0.569	12
Cough	3 (1.7%)	0 (0%)	0.078	3	1 (0.6%)	0 (0%)	0.420	1

^1^ AIRD—autoimmune rheumatic diseases. Statistical significant values are marked with Bold.

**Table 3 jpm-11-01283-t003:** Determinants of the flare-ups (Cox model).

Variable	B	*p*	OR with 95%CI ^1^
More than one immune disease	0.667	0.079	1.9 (0.92–4.09)
Corticotherapy	0.438	0.255	1.5 (0.73–3.3)
Vaccine	−0.630	0.094	0.53 (0.25–1.11)
Flare-up during the previous year	0.972	0.019	2.64 (1.17–5.97)

OR-odds ratio, CI ^1^-confidence interval.

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
