# Peer review of "COVID-19 Vaccine Does Not Increase the Risk of Disease Flare-Ups among Patients with Autoimmune and Immune-Mediated Diseases"

_jpm, 2021, doi:10.3390/jpm11121283_

Round 1
Reviewer 1 Report
This important prospective study conducted by the authors, is attempting to cope with the concern raised by the medical society regarding the potential risk of disease flares in patients with inflammatory/autoimmune diseases following COVID-19 vaccine. The authors concluded that there should not be a reason to concern about the potential risk of flares in patients with inflammatory/autoimmune diseases following COVID-19 vaccine. The study is well designed and the control groups were appropiately used. Minor comments: The Introduction is very short and it ill be helpful for the readers to have more background about what is known in the literature and why there is concern of flares in inflammatory/autoimmune diseases following viral vaccines in general and following COVID-19 vaccine in particular.
Author Response
Thank you for the assessment done on our manuscript and for the opportunity to revise our work. We’ve carefully read your comments and suggestions and prepared an improved version of the manuscript accordingly. Below please find attached a point-by-point response to the issues you raised and the revised document with changes highlighted throughout the manuscript.
Minor comments: The Introduction is very short and it ill be helpful for the readers to have more background about what is known in the literature and why there is concern of flares in inflammatory/autoimmune diseases following viral vaccines in general and following COVID-19 vaccine in particular.
Response: We’ve changed the introduction into:
“The last recommendations published by The European League Against Rheumatism (EULAR) support vaccination in patients with autoimmune and immune mediated diseases (AID-IMD) during the quiescent phase of the disease and before starting immunosuppressive treatment if possible [1]. It is well known that the rate of vaccination is suboptimal in these patients, attributed to low rate of referral for vaccination [2] and the concerns related to disease flare. Until now, epidemiological studies did not support the hypothesis that vaccines may reactivate or may induce AID-IMD [3]. There were reported a few cases of Guillain-Barre syndrome after Influenza vaccine and rare cases of systemic lupus erythematosus exacerbation after papilloma vaccination [4, 5]. Due to the risk of COVID-19 infection, AID-IMD patients were included in the second line of vaccination. However, this decision caused concerns regarding the COVID-19 vaccines safety and effect on disease activity. After the vaccination in large population begun, case series of post-vaccine immune phenomena were published [6, 7]. Disease flare in patients with AID-IMD in clinical remission were reported [8, 9]. Providing reliable study results concerning this subject is important for future research prospects.
The aim of the study was to assess the potential risk of disease flare-up in AID-IMD patients after receiving the COVID-19 vaccine, during a follow-up period of 6 months. ”

Reviewer 2 Report
Dear Editor, I would like to thank for the opportunity this manuscript. This is a succinct and well-presented manuscript that focuses defining the potential risk of disease flare-up in autoimmune/immune mediated diseases patients after receiving the COVID-19 vaccine. Despite limitations, these results may provide an acceptable safety profile for COVID-19 vaccines among patients with autoimmune/immune mediated diseases. Study design and presentation of the results are quite appropriate for the scope of the manuscript. However, there are some issues and questions that should be addressed in the study.
1-It would be valuable to report the distribution of the flares according to the types of COVID-19 vaccines.
2-Please add flares occured after the first or second dose of COVID-19 vaccine to results
3- As the primary outcome measure of the study was the time to first flare, please include more details on the number of days after vaccine when flare started(e.g., 1 day, 1-7 days, and >7 days) and length of flare (e.g., 1 day, 1-7 days, and >7 days) .
4-There are a number of grammatical/spelling errors which need correcting. Some examples include:
- P2 Line 58: Please correct “in study” with “in the study”
- P6 Line 145: Please correct “in accordance to our results”
- P7 Line 182: Please correct “according to our results“ with “according to our results,”
5-Finally, some relevant studies could enrich your manuscript.
-Watad A, De Marco G, Mahajna H, et al. Immune-Mediated Disease Flares or New-Onset Disease in 27 Subjects Following mRNA/DNA SARS-CoV-2 Vaccination. Vaccines (Basel). 2021;9(5):435. Published 2021 Apr 29. doi:10.3390/vaccines9050435
-Barbhaiya M, Levine JM, Bykerk VP, et al. Systemic rheumatic disease flares after SARS-CoV-2 vaccination among rheumatology outpatients in New York City. Annals of the Rheumatic Diseases 2021;80:1352-1354.
-Ishay Y, Kenig A, Tsemach-Toren T, et al. Autoimmune phenomena following SARS-CoV-2 vaccination. Int Immunopharmacol. 2021;99:107970. doi:10.1016/j.intimp.2021.107970
- Terracina KA, Tan FK. Flare of rheumatoid arthritis after COVID-19 vaccination. Lancet Rheumatol. 2021;3(7):e469-e470. doi:10.1016/S2665-9913(21)00108-9
Author Response
Thank you for the assessment done on our manuscript and for the opportunity to revise our work. We’ve carefully read your comments and suggestions and prepared an improved version of the manuscript accordingly. Below please find attached a point-by-point response to the issues you raised and the revised document with changes highlighted throughout the manuscript.
1-It would be valuable to report the distribution of the flares according to the types of COVID-19 vaccines.
Response: We’ve added in the text: “There was no significant difference concerning the types of COVID-19 vaccines administrated in terms of flare development: Pfizer- BioNTech (19/359, 5%), AstraZeneca (4/37, 11%) or Moderna (2/12, 17%), p=0.43.”
2-Please add flares occurred after the first or second dose of COVID-19 vaccine to results.
Response: We’ve added in the text: “Three of the 25 flare-ups were reported after receiving the first dose, while the rest developed after the second dose.”
3- As the primary outcome measure of the study was the time to first flare, please include more details on the number of days after vaccine when flare started(e.g., 1 day, 1-7 days, and >7 days) and length of flare (e.g., 1 day, 1-7 days, and >7 days).
Response: We’ve added in the Results section: “The 3 flares reported after the first dose started on days 8, 10 and 12, respectively. 16% (4/25) of the flares started on day 2, 16% (4/25) between days 14-21 and 56% (14/25) 21 days after the second dose. The length of flare was < 7 days in 30% of the patients, 8-21 days in 22% of the patients and > 30 days in 48% of the patients. However, there was no difference regarding the median length of flare between the two groups: 30 days (4, 149) in the vaccinated and 27.5 days (14, 164) in the non-vaccinated patients, p= 0.803.”
We’ve added in the Discussion section: “Nevertheless, based on the results published until now, it appears that most of the flares usually occurred in the first week following vaccine administration [20, 21], as such post-vaccination long them follow-up might not be required. The fact that almost half of the flares reported in our study developed 21 days after vaccine administration raises the possibility that not all of them were vaccine associated. ”
4-There are a number of grammatical/spelling errors which need correcting. Some examples include:
P2 Line 58: Please correct “in study” with “in the study”
P6 Line 145: Please correct “in accordance to our results”
P7 Line 182: Please correct “according to our results“ with “according to our results,”
Response: We reviewed the article and corrected the spelling and punctuation mistakes
5-Finally, some relevant studies could enrich your manuscript.
-Watad A, De Marco G, Mahajna H, et al. Immune-Mediated Disease Flares or New-Onset Disease in 27 Subjects Following mRNA/DNA SARS-CoV-2 Vaccination. Vaccines (Basel). 2021;9(5):435. Published 2021 Apr 29. doi:10.3390/vaccines9050435
-Barbhaiya M, Levine JM, Bykerk VP, et al. Systemic rheumatic disease flares after SARS-CoV-2 vaccination among rheumatology outpatients in New York City. Annals of the Rheumatic Diseases 2021;80:1352-1354.
-Ishay Y, Kenig A, Tsemach-Toren T, et al. Autoimmune phenomena following SARS-CoV-2 vaccination. Int Immunopharmacol. 2021;99:107970. doi:10.1016/j.intimp.2021.107970
- Terracina KA, Tan FK. Flare of rheumatoid arthritis after COVID-19 vaccination. Lancet Rheumatol. 2021;3(7):e469-e470. doi:10.1016/S2665-9913(21)00108-9
Response: We’ve assessed the suggested references and decided to include all of them in the manuscript.

Round 2
Reviewer 2 Report
I am satisfied that the authors have addressed all of my previous concerns about the article. It is now much improved and I feel that it is now suitable for publication.